# Self-reports of Dutch dog owners on received professional advice, their opinions on castration and behavioural reasons for castrating male dogs

**Pascalle E. M. Roulaux** *, **Ineke R. van Herwijnen** , **Bonne Beerda**

Department of Animal Sciences, Behavioural Ecology Group, Wageningen University and Research, Wageningen, The Netherlands

* pascalleroulaux@hotmail.com

**Data Availability Statement:** All relevant data are within the manuscript and its Supporting Information files.

## Abstract

Male dogs are often castrated based on the thought that it facilitates well-behavedness. However, the causal evidence for this from prospective studies lacks and the existing associative studies present mixed results depending on the studied behaviours. We aimed to gain insight into possible factors driving an owner's decision to castrate their male dog, through a quantitative survey based on a convenience sample. We determined the advice owners received from three types of dog professionals (veterinarian practitioners, behavioural trainers, behavioural therapists) and the owners' assessments of castration's behavioural effects. Data on 491 Dutch owners of castrated and intact male dogs were analysed with Chi-square tests. Results indicate that owners of both castrated and intact dogs received castration advice most often from veterinarian practitioners, with pro-castration at higher frequencies for owners of castrated dogs (Chi-square, P<0.001). Overall, most owners disagreed with or were neutral about statements on castration positively affecting male dog behaviour at a population level. Nevertheless, 58% ($N$ = 145) of the owners of castrated dogs ($N$ = 249) reported that correcting unwanted behaviour was a reason to castrate their own male dog. Unwanted behaviour involved aggression in 50% ($N$ = 70) of the owner-dog dyads. Castrated dog's aggression changes were reported on most as 'no change'. The second most common answer indicated an aggression decrease in dogs castrated to correct unwanted behaviour and an increase in dogs castrated for other reasons (Chi-square, P<0.001). The increase in aggression in a subset of castrated dogs is concerning, as aggression can pose risks to the dog's welfare. We acknowledge the limitations of our study which identifies associations rather than provides causal evidence. Still, we recommend professionals' awareness of possible negative behavioural changes following castration, like increased aggression. Future research on behavioural consequences of castrating dogs needs to build a more solid knowledge base for balanced advice regarding castration.

**Funding:** The author(s) received no specific funding for this work.

**Competing interests:** The authors have declared that no competing interests exist.

## 1. Introduction

Desexing dogs regards the surgical removal of the testes in males, more commonly known as castration, or the ovaria in females. Desexing is a common practice in Western societies. Percentages of 54% of 431 British dogs [1] and of 78% of 413 Australian dogs [2] illustrate how the desexing of dogs is common practice, also in regions where dog reproduction is under control. Dog owners may deem desexing 'the right thing to do', as 74% of 1,016 dog owners considered desexing a practice that their relatives would agree with, 69% reported desexing to be important and 62% expected favourable outcomes of desexing [3]. Thus, the common belief is that desexing makes a dog more well-behaved, but convincing causal scientific evidence for the precise behavioural effects of desexing dogs is presently lacking. This compromises the quality of the advice to dog owners by dog professionals such as veterinarian practitioners, behavioural trainers (also known as dog trainers), and behavioural therapists. These dog professionals ordinarily provide the science-based information that misses in a dog owner's cordial social surroundings [4]. Therefore, we aimed to establish why dog owners decide to castrate their male dog or not and which professional advice, to which avail, is reportedly received by them. Our findings should contribute to a future understanding of how professional advice may support carefully weighed decisions by dog owners on the castration of their dog.

Carefully weighed decisions on castrating dogs are necessary as the scientific information on the effects of desexing dogs, males in particular, is complex and to date incomplete. Strong causal evidence on castration affecting a broad range of behaviour is lacking [5, 6]. Castration of the male dog has been related to reduced inter-male aggression, marking and roaming [7, 8], indicating possible benefits of the procedure. Also, dog bite risk at the population level was higher for intact than desexed dogs [5]. However, effect sizes ranged across the six reviewed studies. Moreover, confounders were considered important to better understand desexing effects on dog behaviour and these confounders regarded for example breed and desexing age, but also environment, such as dog care practices [5]. The complexity increases by findings of desexing associating with unwanted behaviours like fear and types of aggression other than inter-male aggression [9, 10, 11, 12, 13, 14]. The mechanism behind the found associations is that sex hormones are known to have a muting effect on the stress system [15]. For instance, men with higher levels of the sex hormone testosterone have lower levels of pain and fear [16], and for fear this was demonstrated also in male mice [17]. Extrapolation of these findings to male dogs would mean that castration-induced drops in testosterone levels raises fear levels. Since fear is a common motivational factor for aggression in dogs [18, 19, 20], an increase in fear following castration could increase a dog's aggression. Indeed, in a cross-sectional study castrated male dogs ($N = 16$) acted more fearfully and aggressively than intact males ($N = 18$) when interacting with each other during behavioural tests [12]. Indefinite study outcomes that also include female dogs even further complicate the already complex relation between desexing and behaviour. There was no difference in aggression levels between English Cocker Spaniels that were desexed and intact, at least after excluding dogs that were desexed for reasons of correcting unwanted behaviours [21]. Also, no evidence for aggression changes in desexed dogs was found in a large survey-based study on 13,237 to 13,795 dogs [22]. This again after excluding the dogs that were desexed for reasons of correcting unwanted behaviour and when considering multiple aggression affecting factors [22]. With this complexity in findings on desexing and behaviour and the lack of causal evidence [5, 6], how dog owners are advised becomes of interest. This as there is reason to assume that desexing facilitates aggression in a subset of dogs. Aggression is a common burdensome unwanted behaviour. It was diagnosed in 16–72% (depending on the aggression type) of 1,644 dogs included in a retrospective case

series evaluation of medical records [18] and reported by as many as 36% of 174 South Korean dog owners [9]. Also, aggression was the main reason for behavioural consultation in a study population of 140 dogs, of which 129 were desexed [23], and aggression associated with high perceived costs of dog ownership and with reduced ownership satisfaction [24]. Strongly reduced ownership satisfaction due to a dog's unwanted behaviour may ultimately lead to dog relinquishment [25, 26, 27] or euthanasia [28]. The potential dire consequences of unwanted behaviour in dogs make it important that the decision to desex an individual dog is carefully weighed [6]. In this, the advisory role lies logically with professionals such as veterinarian practitioners, behavioural trainers, and behavioural therapists. Today, little is known about which professionals are consulted on this topic by dog owners, what advice these owners receive and how their decision to desex their dog relates to their particular opinions on desexing. These gaps in knowledge underlie the present research and we specifically address the castration of male dogs. We focus on male dogs because in our sample of dog owner reports the desexing for reasons of correcting unwanted behaviour involved males (58%, $N$ = 145 of 249) much more often than females (11%, $N$ = 28 of 258). Desexing may be common in female dogs, like that 83% of Australian female dogs were desexed versus 74% of males [2], but the decision to desex to correct unwanted behaviour seems to involve especially males.

In regions where dog reproduction is under control, dog owners should weigh the benefits and risks of desexing on the basis of balanced and correct information. Dog professionals such as veterinarian practitioners, behavioural trainers, and behavioural therapists may provide the advice that owners need, and we aimed to establish possible factors driving dog owners' decisions to castrate male dogs, including the factors of owner received professional advice, their opinions on castration and behavioural reasons for castrating male dogs. More insight into how owners opinion and decide on castrating their male dog can benefit professional advice, thus allowing owners to make carefully weighed decisions. These decisions ultimately benefit the dog's welfare and the owner-dog relationship.

## 2. Methods

### 2.1. Web-based survey and participant recruitment

A convenience sample of dog owners filled out an internet survey on the desexing of both male dogs and female dogs. Survey items were developed by us, as we were unaware of previous instruments being developed for measuring on the factors of our interest. We pretested the survey with native speakers for understanding and readability. We then analysed the owners' reports in particular for how they were advised by dog professionals on the castration of male dogs, how they opinioned on castration affecting male dog behaviour at a population level and how owners of castrated dogs evaluated the behavioural effects of castration on their own dog specifically. Finally, we studied how satisfied owners of castrated and intact dogs were with having their dog. We targeted dog owners via websites, social media channels and newsletters directed at dog owners. Once posted through these channels, the survey could be shared by dog owners and content managers. We are unaware of studies characterizing the population of Dutch dog owners and could not compare our study sample to information on this population. By gathering and describing demographic characteristics of our study sample, we aimed to provide some insight on the participants to our convenience sample. The survey introduction explained how we considered intact as 'no modification' and desexed as 'surgical removal of testes or ovaria'. We excluded reports on chemically desexed dogs and focussed on irreversible surgical desexing. Also, we excluded reports on dogs that were desexed before they were obtained by their current owner, since such owners had not been involved in the decision to desex and could not assess the behavioural changes following desexing.

The survey consisted of a first part and a second part that varied in length, depending on the dog being intact or desexed (see S1 Appendix for survey items). The first part assessed characteristics of the owner and dog, being the owner's gender, education level and age and the dog's sex, current age, age at acquisition by the current owner, breed, pedigree, desexing status, and age at the moment of desexing. Furthermore, owners indicated the percentage of time that they take care of the dog for which they filled in the survey, and we excluded owners who indicated that they take care of the dog less than 50% of the time. This part of the survey also held questions on the owners' opinion on the behavioural effects of desexing and owners scored their general dog ownership satisfaction as 'not at all satisfied', 'not very satisfied', 'moderately satisfied', 'satisfied' and 'very satisfied'. Dog ownership satisfaction scores were skewed towards (very) satisfied and the original five-point scale was expressed binary, with 1 being very satisfied and 0 being less than very satisfied. Owners expressed their opinion on how desexing affects male dog behaviour at population level, so not the behaviour of their own dog specifically, by indicating their (dis)agreement with statements on the favourable effects of desexing. Statements were on the behaviours of aggression, calmness, dog-directed sociality, dominance, human-directed sociality, mounting, roaming, trainability and urine marking, and were for instance 'desexing diminishes aggression in male dogs' and 'desexing makes male dogs calmer'. Participants could answer by selecting 'strongly disagree', 'slightly disagree', 'neutral', 'slightly agree' or 'strongly agree'.

In the second part of the survey, both the owners of intact dogs and desexed dogs reported whether or not they had received advice from the different dog professionals (i.e. veterinary practitioners, behavioural trainers, and behavioural therapists) concerning the desexing of their dog. They reported on received advice as 'not in favour of desexing', 'neutral' and 'in favour of desexing', which hereafter we refer to as 'con-castration', 'neutral' and 'pro-castration'. Owners of desexed dogs scored reasons for having their dog desexed on a five-point Likert scale from 'not relevant as a reason for desexing' to 'the main reason for desexing'. The reason 'to correct unwanted behaviour' was analysed in more detail and expressed as a binary number with 1 meaning the correction of unwanted behaviour had played a role in the decision to desex, varying from a small role to main reason, and 0 meaning it had not played any role. Owners who had their dog desexed to correct unwanted behaviour were divided binary with the number 1 indicating that aggression was a behavioural problem to correct, and 0 meaning it had not played any role. These binary scores were derived from a four-point Likert scale that ranged from 'not problematic' to 'the main problem to be corrected by desexing'. The owners who had their dog desexed to correct unwanted behaviour indicated their satisfaction with the behavioural effects of desexing on a scale from 'completely dissatisfied', 'slightly dissatisfied', 'neutral', 'largely satisfied' to 'completely satisfied'. Lastly, all owners of desexed dogs, regardless of their reason for desexing, reported how desexing had changed the prevalence of aggression. The original five-point scale of 'strongly decreased', 'slightly decreased', 'no change', 'slightly increased' and 'strongly increased' was converted to the three-point scale of 'decreased', 'unchanged' and 'increased' for further analyses.

## 2.2. Data processing and statistical analyses

From our full sample we excluded the participants who did not indicate their dog's sex (male or female) and/or status (desexed or intact) and/or applicability of desexing as correction of behaviour in their dog and/or taking care of the dog at least 50% of the time. From the remaining sample of 1,006 owner reports we excluded those that involved dogs that were chemically desexed ($N = 46$), desexed before acquisition by the current owner ($N = 95$), or female (desexed $N = 258$, intact $N = 116$). Correcting unwanted behaviour had played a role in the decision to

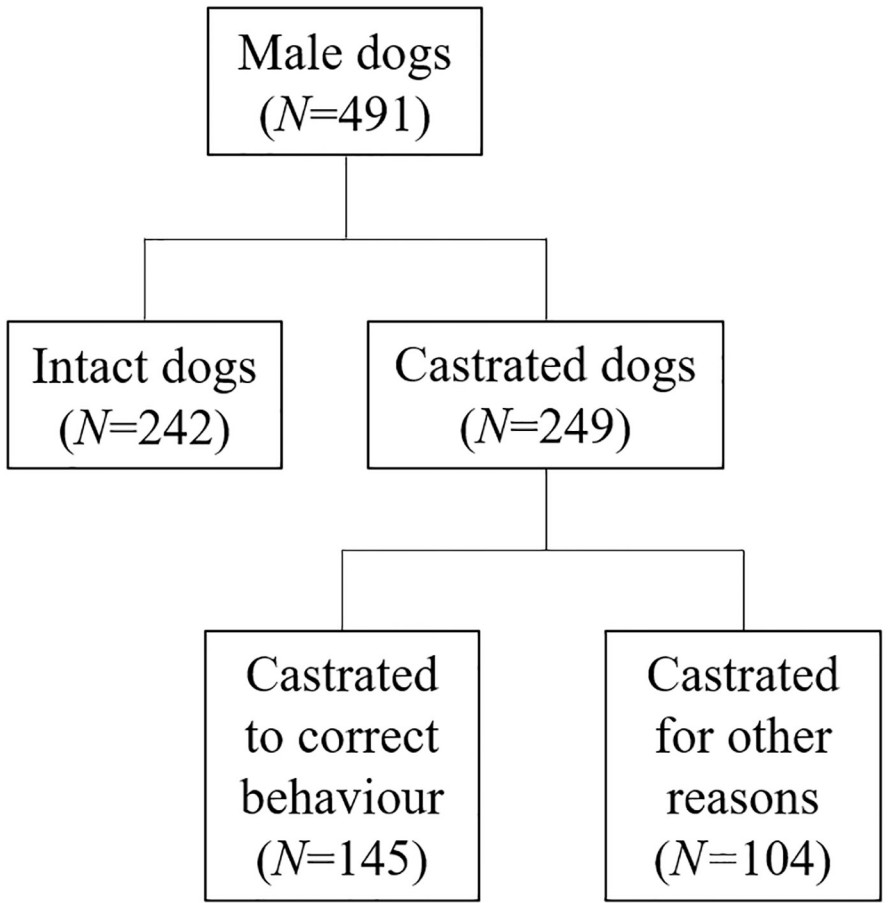

**Fig 1. Subsamples within the sample of 491 owners of male dogs.** Dogs were either castrated ($N$ = 249) or intact ($N$ = 242). For castrated dogs, the owners indicated whether or not correcting unwanted behaviour had played a role in the decision to castrate, thus we divided the castrated dogs into 'castrated to correct behaviour' ($N$ = 145) and 'castrated for other reasons' ($N$ = 104).

desex in only 11% ($N$ = 28) of desexed females, but in the majority of desexed males (58%, $N$ = 145). Thus, for this study we used the data on 491 male dogs only (castrated $N$ = 249, intact $N$ = 242; see Fig 1 for subsample details). The group of owners of castrated male dogs subdivided into a group of 'owners who had their dog castrated to correct unwanted behaviour' ($N$ = 145) and a group of 'owners who had their dog castrated for other reasons only' ($N$ = 104). The online survey allowed that the occasional question was left unanswered and the precise sample sizes used for analyses are given in the results section.

Based on the lowest subsample size of 104 (performing Chi-square tests of df = 2), the statistical power was 0.79 for detecting at least medium effect sizes of 0.30, while maintaining an α-level of 0.05. This power was calculated using the Chi-square power calculator of the https://www.masc.org.au/. We did not set a predetermined sample size but arbitrarily aimed to detect at least medium effect sizes ($\sqrt{(\chi 2/n)} > 0.30$). With a statistical power of at least 0.8 [29, 30]. We tested for pairwise differences between the frequency and the nature of advice that was received from the three types of professionals, and we tested how the frequency and the nature of this advice differed between reports on castrated and intact dogs. Opinions on the effects of castration on the behaviour of male dogs at a population level were compared, also between

owners of intact and castrated males. Owner perceived changes in dog aggression following castration were compared between owners who had their dog castrated to correct unwanted behaviour and owners who had their dog castrated for other reasons. Finally, general satisfaction with dog ownership was compared between owners of intact dogs, owners of dogs that were castrated to correct unwanted behaviour and owners of dogs that were castrated for other reasons. All these comparisons were made with Pearson's Chi-square tests, maintaining a level of significance of P<0.05 and P-values are reported throughout. Chi-square values and residuals and the degrees of freedom for each Chi-square test are presented in S2 Appendix.

### 2.3. Ethical statement

The online survey's introduction explained the purpose of the research and the study did not involve treatments or interventions in the life of participants or their dogs. The survey was not repeated, meaning it did not interfere significantly with normal daily life, and did not include questions that were psychologically burdening. This exempts the study from review by our ethics committee, according to the guidelines of Wageningen University Medical Ethics Review Committee (Medisch Ethische Toetsingscommissie van Wageningen University, METC-WU). Informed consent was not obtained as participants chose to participate freely via internet and the purpose of the research was stated at the start of the online survey.

## 3. Results

### 3.1. Participants and their male dogs

The participants ($N = 491$) were selected from the larger sample of 1,006 records as detailed in the Methods section. Typically, participants were women (89%, $N = 433$; men: 11%, $N = 53$) and over 34 years of age (<35 years: 30%, $N = 146$; 35–44 years: 17%, $N = 84$; 45–54 years: 33%, $N = 160$; >54 years: 21%, $N = 101$). The majority had completed higher professional education (52%, $N = 255$; vocational education above high school level: 33%, $N = 162$; below this level: 15%, $N = 73$). The majority of the male dogs were obtained by the owner before they were 10 weeks old (67%, $N = 329$). The remainder was obtained between 10 weeks and 1-year-old (24%, $N = 116$; >1-year-old: 9%, $N = 44$). Dogs were of various breeds and about half of them were certified pedigree dogs (54%, $N = 262$; 46% look-a-like or mixed breed, $N = 227$). The majority of dogs was aged between 1 and 8 years old (<1 year: 6%, $N = 27$; 1–2 years: 15%, $N = 74$; 2–4 years: 26%, $N = 127$; 4–6 years: 20%, $N = 99$; 6–8 years: 14%, $N = 69$; >8 years: 19%, $N = 94$). Reproductive status was near evenly distributed in our sample, with 51% ($N = 249$) being castrated (intact, $N = 242$). Of all castrated dogs, 27% ($N = 68$) was castrated before they were 1 year old, 40% ($N = 99$) was between 1 and 2 years old and 32% ($N = 81$) was over 2 years old at the time of castration.

### 3.2. Professional advice on castrating male dogs

Owners ($N = 491$) had received professional advice on the castration of their own dog most often from veterinarian practitioners (72%, $N = 347$), followed by behavioural trainers (48%, $N = 224$) and behavioural therapists (38%, $N = 174$). All three pairwise differences were significant (Chi-square tests, all P-values <0.001; for details see Table A in S2 Appendix). Of all owners who received advice ($N = 380$), 37% ($N = 142$) received advice from only one type of professional, 29% ($N = 111$) received advice from two types and 33% ($N = 127$) received advice from all three types of professionals.

We assessed how professional advice that was received by each individual owner related to the owner's decision to castrate their own dog, by comparing the frequency of received

**Table 1. Frequencies of advice on castrating male dogs as received from three types of dog professionals by owners of intact and castrated male dogs.**

| | Owners of intact dogs | | | Owners of castrated dogs | | | |
| --- | --- | --- | --- | --- | --- | --- | --- |
| | Not advised | Advised | Total | Not advised | Advised | Total | P-value |
| Veterinarian | 38% (N = 90) | 63% (N = 150) | 240 | 19% (N = 47) | 81% (N = 197) | 244 | <0.001 |
| Trainer | 53% (N = 122) | 47% (N = 110) | 232 | 50% (N = 116) | 50% (N = 114) | 230 | 0.644 |
| Therapist | 63% (N = 147) | 37% (N = 86) | 233 | 61% (N = 140) | 39% (N = 88) | 228 | 0.709 |

Advice from dog professionals (veterinarian practitioners, behavioural trainers, and behavioural therapists) on the castration of male dogs was reported by dog owners (N = 491). We compare percentages of advised versus not advised between owners of intact (N = 242) and castrated (N = 249) male dogs in three Chi-square tests, one for each type of professional, and thus present P-values per professional. Subsample counts are between brackets and further details are presented in Table B in S2 Appendix.

professional advice between owners of castrated dogs (N = 249) and owners of intact dogs (N = 242) and this for each type of professional (veterinarian practitioner, behavioural trainer and behavioural therapist). Owners of castrated dogs more often than owners of intact dogs received advice from veterinarian practitioners, as opposed to not having been advised by them (81% versus 63%; P<0.001; Table 1 and for details see Table B in S2 Appendix). This difference between owners of castrated and intact dogs was not found in advice that was received from behavioural trainers (P = 0.644) or behavioural therapists (P = 0.709).

In addition to the advice frequency, we assessed the nature of the advice that dog owners received, in terms of it being con-castration, neutral or pro-castration. Veterinarian practitioners had advised in favour of castration most often (44%, N = 171; Chi-square P = 0.005 for veterinarian practitioners versus behavioural trainers; P<0.001 for veterinarian practitioners versus behavioural therapists; for details see Table C in S2 Appendix), followed by behavioural trainers (40%, N = 89) and behavioural therapists (32%, N = 55; P = 0.028 for behavioural trainers versus behavioural therapists).

We then assessed how the nature of professional advice that was received by each individual owner related to the owner's decision to castrate their own dog, by comparing the nature of received professional advice between owners of castrated dogs (N = 249) and owners of intact dogs (N = 242). We found that owners of castrated dogs more often than owners of intact dogs received pro-castration advice from all three types of professionals (Chi-square tests, all P-values <0.001; Table 2 and for details see Table D in S2 Appendix). The percentage of owners of castrated dogs who had received pro-castration advice from veterinarians was more than twofold higher than the percentage of owners of intact dogs who had received this advice, and more than three-fold higher for advice from behavioural trainers and behavioural therapists.

**Table 2. Nature of advice on castrating male dogs as received from three types of dog professionals by owners of intact and castrated male dogs.**

| | Owners of intact dogs | | | | Owners of castrated dogs | | | | |
| --- | --- | --- | --- | --- | --- | --- | --- | --- | --- |
| | Pro | Neutral | Con | Total | Pro | Neutral | Con | Total | P-value |
| Veterinarian | 29% (N = 44) | 36% (N = 54) | 35% (N = 52) | 150 | 64% (N = 127) | 32% (N = 63) | 4% (N = 7) | 197 | <0.001 |
| Trainer | 18% (N = 20) | 35% (N = 39) | 46% (N = 51) | 110 | 61% (N = 69) | 29% (N = 33) | 11% (N = 12) | 114 | <0.001 |
| Therapist | 15% (N = 13) | 23% (N = 20) | 62% (N = 53) | 86 | 48% (N = 42) | 32% (N = 28) | 20% (N = 18) | 88 | <0.001 |

Advice from dog professionals (veterinarian practitioners, behavioural trainers, and behavioural therapists) on the castration of male dogs was reported by dog owners (N = 491). We compare percentages on the nature of advice being in favour of castration (pro), neutral, or against it (con) between owners of intact (N = 242) and castrated (N = 249) male dogs in three Chi-square tests, one for each type of professional, and thus present P-values per professional. Subsample counts are between brackets and further details are presented in Table D in S2 Appendix.

**Table 3. Dog owner opinions on the effects of castration on male dog behaviour at a population level.**

| | Owners of intact dogs | | | | Owners of castrated dogs | | | | |
|---|---|---|---|---|---|---|---|---|---|
| | **Disagree** | **Neutral** | **Agree** | **Total** | **Disagree** | **Neutral** | **Agree** | **Total** | **P-value** |
| Trainability | 55% (N = 111) | 42% (N = 85) | 2% (N = 5) | 201 | 40% (N = 61) | 50% (N = 77) | 10% (N = 15) | 153 | 0.001 |
| Mounting | 48% (N = 94) | 45% (N = 90) | 8% (N = 15) | 199 | 29% (N = 43) | 59% (N = 89) | 12% (N = 18) | 150 | 0.002 |
| Aggression | 57% (N = 112) | 40% (N = 80) | 3% (N = 6) | 198 | 44% (N = 67) | 45% (N = 68) | 11% (N = 16) | 151 | 0.005 |
| Sociality$_{human}$ | 61% (N = 121) | 38% (N = 75) | 2% (N = 4) | 200 | 45% (N = 69) | 50% (N = 76) | 5% (N = 7) | 152 | 0.013 |
| Sociality$_{dog}$ | 57% (N = 114) | 38% (N = 76) | 5% (N = 9) | 199 | 44% (N = 67) | 46% (N = 71) | 10% (N = 15) | 153 | 0.018 |
| Calm | 47% (N = 94) | 42% (N = 84) | 11% (N = 23) | 201 | 36% (N = 56) | 46% (N = 72) | 18% (N = 29) | 157 | 0.052 |
| Marking | 47% (N = 94) | 43% (N = 85) | 10% (N = 20) | 199 | 37% (N = 56) | 49% (N = 74) | 14% (N = 22) | 152 | 0.118 |
| Roaming | 33% (N = 65) | 48% (N = 96) | 20% (N = 39) | 200 | 24% (N = 36) | 55% (N = 84) | 22% (N = 33) | 153 | 0.180 |
| Dominance | 53% (N = 105) | 40% (N = 80) | 7% (N = 14) | 199 | 45% (N = 68) | 45% (N = 68) | 11% (N = 16) | 152 | 0.250 |

Dog owners reported their opinion on the behavioural effects of castration in male dogs at a population level, so not specifically for their own dog. Their opinions, categorized as disagree (with favourable effects), neutral and agree, were on nine different behaviours and percentages are given separately for owners of intact dogs (N = 242) and castrated dogs (N = 249). Chi-square P-values are presented per behaviour and further details are presented in Table E in S2 Appendix.

### 3.3. Owners' opinions on the effects of castration on male dog behaviour at a population level

The owners of male dogs (N = 491) reported on how they believe castration affects the behaviour of male dogs at a population level, so not specifically the behaviour of their own dog, by rating their (dis)agreement with presumed favourable effects on nine behaviours. We combined their ratings on a five-point scale into the categories of disagreement, neutral and agreement. Owners more often disagreed with the presumed favourable effects of castration (45% of owners across the nine behaviours) than that they agreed (9%), which was more pronounced in the group of owners who owned an intact dog (51% disagreed, 8% agreed) than in the group of owners who owned a castrated dog (38% disagreed, 12% agreed). This difference was significant for trainability (P = 0.001; Table 3 and for details see Table E in S2 Appendix), mounting (P = 0.002), aggression (P = 0.005), human-directed sociality (P = 0.013) and dog-directed sociality (P = 0.018), and there was a trend for calmness (P = 0.052).

### 3.4. Changes in aggression following castration of the own male dog

Owners of castrated male dogs also reported on changes in their own dog's behaviour following castration and we were interested in changes in aggression. Generally, more than half of the owners (58%, 145 out of 249) indicated that correction of unwanted behaviour had played a role in their decision to have their dog castrated, varying from it being a side issue to the main reason. For half of these owners (70 out of 140; five missing values) the unwanted behaviour of aggression was of concern, varying from it being somewhat problematic to the main problem behaviour to correct. All owners of castrated male dogs, regardless of the reason for castration, then reported how they evaluated the changes in aggression levels in their dogs after castration, which we categorized as decreased (strongly or slightly on the original five-point scale), unchanged or increased (slightly or strongly). This question held a large number of missing values (29%, N = 72 out of 249). A decrease in aggression after castration was reported by 32% (N = 56 out of 177) of the participants who answered this question; unchanged by 51% (N = 90 out of 177) and an increase was reported by 18% (N = 31 out of 177). In more detail, we compared the owners who had their dog castrated for reasons of correcting behaviour and those who had their dog castrated for other reasons. In this comparison,

owners who had their dog castrated for reasons of correcting behaviour reported to a higher degree that aggression decreased (42%, *N* = 48 out of 114) than owners who had their dog castrated for other reasons (13%, *N* = 8 out of 63; P<0.001; for details see Table F in S2 Appendix) and reported to a lower degree no changes (43%, *N* = 49 out of 114 versus 65%, *N* = 41 out of 63).

### 3.5. Owner satisfaction

We tested the owners' satisfaction with castration for reasons of correcting their dog's behaviour and we tested general ownership satisfaction for all owners. The owners who had their dog castrated for reasons of correcting its behaviour were mostly satisfied with how castration had affected the behaviour(s) of concern. Forty-seven percent (*N* = 65 out of 137) was completely satisfied and 25% (*N* = 34) was largely satisfied. Others were neutral about the effects (1%, *N* = 1), slightly dissatisfied (7%, *N* = 9) or completely dissatisfied (20%, *N* = 28).

General dog ownership satisfaction was expressed as a binary score of very satisfied or less than this and we compared owners of intact dogs (*N* = 242) with owners of dogs that were castrated to correct behaviour (*N* = 145) or castrated for other reasons (*N* = 104). Of the owners whose dogs were castrated to correct behaviour, 53% reported being very satisfied with having their dog (*N* = 76 out of 144). This was significantly less than owners of intact dogs (69%, *N* = 167 out of 241; P = 0.001) and owners of dogs that were castrated for other reasons (69%, *N* = 72 out of 104; P = 0.009) reported being very satisfied (Chi square tests; for details see Table G in S2 Appendix).

## 4. Discussion

Desexing dogs is a common practice [1, 2] that according to popular belief is 'good for the dog's behaviour' [3], apparently especially when it concerns male dogs. We wanted to better understand how dog owners' decisions to desex their dogs are influenced by professional advice and the common beliefs about the behavioural consequences. This is especially important as presently prospective causal data in this area are missing [5, 6], but individually based advice is recommended in areas where population management does not apply [6, 31]. Consequently, advising professionals such as veterinarian practitioners, behavioural trainers and behavioural therapists could benefit from information on how dog owners decide to castrate their male dog or not.

Our sample of mainly female owners of various dog breeds had a near equal spread between pedigree and non-pedigree dogs and more importantly, a near even spread on reproductive status. Owners in this particular sample reported that they received professional advice on castration most often from veterinarian practitioners (72%, behavioural trainers: 48%, behavioural therapists: 38%). Owners did not indicate the reasoning behind their professionals' advice, and this could be based on non-behavioural reasons. Regardless of the reason, advice received from veterinarian practitioners was more often pro-castration (49%) than con-castration (17%) in which they opposed behavioural therapists (pro-castration: 32%, con-castration: 41%). Furthermore, receiving advice from veterinarian practitioners was reported more often by the owners of castrated dogs than by the owners of intact dogs and for owners of castrated dogs it had been more often pro-castration. Nevertheless, owners of both castrated and intact dogs reported low levels of agreement with statements on castration having favourable effects on the behaviour of male dogs at a population level, though owners of castrated dogs agreed one and a half times more often than did owners of intact dogs. These overall low levels of agreement seemingly contradict that in 58% of the castrated dogs correcting behaviour had played a role in the decision to castrate. These 58% of owners reported unchanged or decreased levels of aggression following castration, whereas owners who castrated their dog

for other reasons most often reported aggression to be unchanged or increased. Finally, our results indicate that overall satisfaction with dog ownership was higher for owners of intact dogs than for owners of dogs castrated to correct behaviour.

Veterinarian practitioners apparently are an important driver of the owner's decision to castrate, as they advise owners often and pro-castration. As most veterinarian practitioners are professionals in veterinary care more than in behaviour, this finding merits a warning against a Dunning-Kruger effect, when veterinarian practitioners advise on the likely behavioural outcomes of a male dog's castration. The Dunning-Kruger effect is a psychological effect that implies an overestimation of competence in areas where one lacks competence [32]. As an example in another advising profession, the Dunning-Kruger effect showed in a group of 94 volleyball coaches who advised high schoolers on the volleyball court. These coaches were compared for their self-reported efficacy scores for coaching abilities. Coaches in the lowest quartile of coaching abilities reported significantly higher efficacy than coaches in the highest quartile of coaching abilities [33]. The Dunning-Kruger effect has been found in a wide range of disciplines and professions [34, 35], and was suggested to affect veterinarian students as well [36]. Particularly as sound scientific data on the behavioural effects of castration are presently lacking [5, 6], the Dunning-Kruger effect could imply an overestimation of competence to advise on behavioural effects of castration and may cause veterinarian practitioners to unintentionally advise too confidently on castration affecting male dog behaviour favourably.

Owners could also be vulnerable to psychological processes affecting their opinions on castration. One such process is cognitive dissonance, which facilitates an individual's coping with its environment, by aligning information processing and decision taking [37]. It prevents discomfort, by processing information selectively [38, 39]. Namely, information that is in line with held cognitions is processed, but contradicting information is disregarded. For owners who castrated their dog, noticing advantageous outcomes of castration may be less discomforting than noticing disadvantageous outcomes, which would not be in line with the owners' cognitions. Disadvantageous outcomes that are in discord with the received professional advice could add to the dissonance. Such cognitive dissonance processes could work along the lines of similar psychological processes noted in owners of brachycephalic ('flat-faced') obstructive airway syndrome (BOAS)-affected dogs. Over half (58%) of these owners reported that their dog did not have a breathing problem [40], although these dogs were affected and consequently showed symptoms. Also, when buying a new dog, owner-perceived dog health was of lesser importance to owners of brachycephalic breeds than non-brachycephalic breeds [41]. Apparently, these owners recognise their dog's health issues insufficiently, as it may be discomforting to the owner to realise that their choice of a dog's appearance affects its welfare negatively. Similarly, if dog owners expect aggression levels to lower after castration, a castrated dog aggressing *more* would cause psychological discomfort, or dissonance. Noteworthy is the apparent contradiction in how owners opinion on castration affecting aggression levels in male dogs at a population level versus in their own dog. Presumed favourable effects are not reported at a population level, but favourable behavioural effects are reported for owned male dogs that are castrated for behavioural reasons. Logically, this could be due to an actual effect of castration on the dogs' aggression levels. However, without prospective and observational data, the process of cognitive dissonance cannot be ruled out. To avoid dissonance, owners could be more receptive to information in line with expectations, meaning that especially actions of the dog that are non-aggressive will be registered, remembered and reported. The process of cognitive dissonance may contribute to the common belief that castration is 'good for the dog's behaviour', although this belief presently is unsupported. Owners of castrated dogs could report positive behavioural effects following castration to other dog owners, or back to their advising professional. The resulting unrealistic public optimism about the

consequences of castrating a male dog may affect a dog owner's decision to castrate and this may unintentionally backfire if for instance fear-induced aggression levels rise. Such unwanted behaviour can lead to dissatisfaction with dog ownership, which may compromise the dog's welfare through increased risk of shelter relinquishment or even euthanasia [25, 26, 27, 42, 43].

Limitations of our study, such as it being based on a convenience sample using online recruitment, imply that causality could not be studied and that our findings likely do not apply to all dog owners, such as becoming clear from our sample including 89% women. Women are often reported to respond at higher levels to survey-based studies, such as seen in a Finnish study on the topic of social class inequalities and health [44] and in a USA-study on non-response in student surveys [45]. For dog studies in particular, a higher percentage of women participants is common also, such as seen in the 93% females of 3,080 participants, with recruitment done online, predominantly using social media [46] and the 91% females of 653 participants, with recruitment done via internet and advertisement cards placed in veterinary hospitals, grooming shops, retailers, etc. [47]. This matters as views on a dog's (and cat's) desexing differed between men and women, with men being less likely to be pro-desexing [31]. Our convenience sample was largely gathered via internet, including social media, and these channels were hypothesized to be operated differently by women and men, such as by women using these more for communicating and men more for information gathering (searching) [48, 49].

Even though our findings on this convenience sample should not be extrapolated to the general population of dog owners without taking into account the limitations of our study, it seems that owners of intact and castrated dogs differ in the advice that they received from professionals, as well as in their opinions on the behavioural consequences of castration. Particularly, we underline the need for gathering causal evidence for the behavioural effects of castration through prospective study set-ups as to facilitate optimal advice to owners. Today, population control arguments are less valid grounds for a male dog's castration in many parts of the world. Several authors stress the importance of individually based advice for owner-dog combinations [6, 31] and only through strong scientific evidence on pros and cons of castration can professionals optimize the provision of such individual advice. Without causal and objective evidence, the common belief that castration is 'good for the dog's behaviour' is an urban legend that spreads readily as it elicits an emotional response, seems plausible and contains practical information or a social moral [50].

We recommend prospective causal study set-ups, rather than the predominantly cross-sectional set-ups that have been adopted so far [9, 10, 11, 12, 13, 14, 21, 22] to upgrade the present knowledge on the behavioural effects of castration in dogs from associative to causal. For veterinarian practitioners particularly, attention to behavioural aspects of castration is recommended. They seem to be consulted most often on the topic of castrating male dogs and may be inclined to overestimate their own competence to advise on behavioural effects of castration, following the psychological processes such as the Dunning-Kruger effect. Particularly in those regions where a dog's castration is not necessary for population management, careful consideration of the pros and cons of castration for an individual dog regarding its health *and* behaviour may benefit the dog's behaviour, relationship with its owner and welfare.

## Supporting information

**S1 Appendix. Survey items.** For this survey-based research we determined the advice owners received from three types of dog professionals (veterinarian practitioners, behavioural therapists, behavioural trainers) and owners' assessments of castration's behavioural effects and this appendix lists survey items in English.
(DOCX)

**S2 Appendix. Tables containing detailed Chi-square test output.** We present additional Chi-square test output, including Chi-square values, degrees of freedom and residuals for all performed analyses. In Tables A and C we compare the frequency and the nature of the advice that was received by our complete sample of dog owners ($N = 491$) between different types of professionals (pairwise comparisons between veterinarian practitioners, behavioural trainers, and behavioural therapists). In Tables B and D we compare the frequency and the nature of the received advice between owners of intact dogs ($N = 242$) and owners of castrated dogs ($N = 249$), and the same two subsamples are used to compare the owners' opinions in Table E. In Table F we compare owner-reported changes in aggression between dogs that were castrated for behavioural correction ($N = 145$) and dogs that were castrated for other reasons ($N = 104$). In Table G we compare the general ownership satisfaction of owners of intact dogs ($N = 242$), owners of dogs that were castrated for behavioural correction ($N = 145$) and owners of dogs that were castrated for other reasons ($N = 104$) in pairwise comparisons. In all Tables, each row represents one Chi-square test. The first column of each table contains the Chi-square value and the degrees of freedom, the last column contains the P-value. Counts represent numbers of dog owners and Chi-square residuals are between brackets and identify significant deviations from expected values (i.e. $> |2|$, in bold).
(DOCX)

**S1 Data.**
(XLSX)

# Author Contributions

**Conceptualization:** Pascalle E. M. Roulaux.

**Data curation:** Bonne Beerda.

**Formal analysis:** Pascalle E. M. Roulaux, Bonne Beerda.

**Methodology:** Pascalle E. M. Roulaux, Bonne Beerda.

**Project administration:** Pascalle E. M. Roulaux.

**Supervision:** Ineke R. van Herwijnen, Bonne Beerda.

**Visualization:** Pascalle E. M. Roulaux.

**Writing – original draft:** Pascalle E. M. Roulaux, Ineke R. van Herwijnen.

**Writing – review & editing:** Pascalle E. M. Roulaux, Ineke R. van Herwijnen, Bonne Beerda.

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
