## [Decision Letter · Decision Letter 0]

13 Feb 2020

PONE-D-20-00127

Possible factors driving dog owners' decision to castrate male dogs - Self-reports of Dutch dog owners on received professional advice, their opinions on castration and behavioural reasons for castrating male dogs

PLOS ONE

Dear Dr Roulaux

Thank you for submitting your manuscript to PLOS ONE. After careful consideration, we feel that it has merit but does not fully meet PLOS ONE’s publication criteria as it currently stands. Therefore, we invite you to submit a revised version of the manuscript that addresses the points raised during the review process.

Many thanks for submitting your manuscript to PLOS One

Your manuscript was reviewed by three experts in the field

All three see merit in the work and they have recommended that some changes be made to the manuscript.

If you could write a detailed response to reviewers, that will speed up the second review

I wish you the best of luck with your revisions

Many thanks

Simon

We would appreciate receiving your revised manuscript by Mar 29 2020 11:59PM. To enhance the reproducibility of your results, we recommend that if applicable you deposit your laboratory protocols in protocols.io, where a protocol can be assigned its own identifier (DOI) such that it can be cited independently in the future. For instructions see: http://journals.plos.org/plosone/s/submission-guidelines#loc-laboratory-protocols

We look forward to receiving your revised manuscript.

Kind regards,

Simon Russell Clegg, PhD

Academic Editor

PLOS ONE

1. We note that you have indicated that data from this study are available upon request. PLOS only allows data to be available upon request if there are legal or ethical restrictions on sharing data publicly. For more information on unacceptable data access restrictions, please see http://journals.plos.org/plosone/s/data-availability#loc-unacceptable-data-access-restrictions.

Reviewers' comments:

Reviewer's Responses to Questions

**Comments to the Author**

1. Is the manuscript technically sound, and do the data support the conclusions?

Reviewer #1: Partly

Reviewer #2: No

Reviewer #3: Yes

2. Has the statistical analysis been performed appropriately and rigorously? 

Reviewer #1: No

Reviewer #2: No

Reviewer #3: Yes

3. Have the authors made all data underlying the findings in their manuscript fully available?

Reviewer #1: No

Reviewer #2: No

Reviewer #3: No

4. Is the manuscript presented in an intelligible fashion and written in standard English?

Reviewer #1: Yes

Reviewer #2: Yes

Reviewer #3: No

5. Review Comments to the Author

Reviewer #1: This is an interesting paper examining the reasons for choosing castration of male companion dogs and its perceived behavioural consequences. The work seems sound but it was problematic not to have sight of the questionnaire, and the statistical approach is very simple and limited. See specific comments below. The paper is generally well written but there are some minor issues with grammar in several places (e.g. “Assumingly,”) – the manuscript would benefit from review from a native English speaker.

Introduction

The specific knowledge gap filled by this study could be clearer. The claim that we do not know the behavioural effects of castration is contradicted by studies cited which examine this.

“why dog owners decide to (not) castrate their male dog” is not helpful wording, suggest “why dog owners decide to castrate their male dog or not”

Methods

Please include a copy of the questionnaire as an appendix or supplementary material – this is normal practice in studies of this type. Currently the reader cannot see the range of reasons presented for castration, for example, which makes review of the paper difficult.

The methods do not describe the collection of any demographic data, yet this data appears in results. Again, we need to see the questionnaire and all aspects of the questionnaire need to be described in the methods.

The statistics are rather simple and very limited. For example, use of Chi square required the collapsing of five point Likert scales into binary categories which seems a shame and represents a loss of detail in the data. By assigning a score based on the Likert scale, many more multivariate analysis options are available which are much more appropriate. Currently no effects of demographics are included in the analysis – then why collect this data? The authors should strongly consider complete re-analysis of their data with more sophisticated techniques, or justify why this is not done.

How were the 491 analysed selected from the total of male dogs? Did this introduce bias?

Discussion

The discussion covers a lot of relevant ground but has two major omissions

(1) Possible bias in the study sample based on online self-selection – this is a major issue with online recruitment

(2) Effects of demographics (which were not statistically analysed)

See: Wonsaengchan, C. and McKeegan D.E.F. (2019). The views of the UK public towards routine neutering of dogs and cats. Animals 9(4): 138, DOI:10.3390/ani9040138

Reviewer #2: The manuscript tackles an important issue. There are many concerns regarding the manuscript.

1) The "introduction" section emphasizes that desexing has predominantly no effect on aggression behavior but misses recent review studies that indicate a beneficial effect of desexing of male dogs in reducing the risk of dog bites and male dog-directed aggression (D'Onise et al., Inj Prev, 2017; Urfer & Kaeberlein, Animals (Basel), 2019).

2) There is neither information on sample size calculation nor on the power to detect differences with the sample obtained.

3) There is no information on how the questionnaire was developed. Is it based on a previously validated instrument? With no specific information on the validity or reliability of the questionnaire, the results are questionable.

4) This is a convenience sample and may not reflect the profile of the owners of dogs in the region. Therefore, with no information on how the sample matches the profile of the population of dog owners, the results might well be biased.

5) Statistical analysis is poorly described and conducted. For instance, there is no need to present Chi-square residuals (tables) we need to know, at least, the p-values associated with each comparison.

6) The presentation of results is confusing and may not reflect actual data. For instance, from table 3, I can see that 10% of owners of castrated dogs agree that castration may have affected aggressive behavior against only 3% of owners of non-castrated dogs. So, beyond the fact that we have almost 30% of missing data for that question and that most owners do not agree, it seems that castration increased the likelihood of a "positive" response (from 3% to 10%), indicating that the experience of castration may have changed the perception of the owner regarding the issue.

7) In any case, it is not very easy to compare different people with different experiences. I may well have not castrated my dogs because my dog is not aggressive or because I simply do not believe in castration for such a purpose. In a counterfactual world, what would be my response if otherwise my dog has been castrated independently of my previous opinion? Of course, this is not represented by the answers of owners of castrated dogs, because these owners might well have decided to castrated their dogs because their dogs were much more aggressive than those of non-castrated dogs. So, comparisons here are not adequate for any causal inference.

8) Results sometimes are interpreted against the facts. For instance, from Table 4, I would say that, compared to dogs castrated for other reasons, dogs castrated for behavioral correction showed a substantial decrease in putative behavior. However, authors prefer to emphasize that "Castrated dog's aggression changes were reported on most as 'no change'."

9) The emphasis on "increased" aggression behavior is not rooted in the data. Increased aggression was the smaller change observed and may reflect just variation of owner perception since it is not based on any objective measure.

Reviewer #3: The article "Possible factors driving dog owners' decision to castrate male dogs - Self-reports of Dutch dog owners on received professional advice, their opinions on castration and behavioural reasons for castrating male dogs" is of interest but some points need to be properly treated to improve the quality of the manuscript.

-The text (specially introduction and discussion) is very extensive and convoluted. It must be fully improved.

-Tables are not properly formatted. They are not self-explanatory and do not contain essential information (P-values and proportions).

-The results of the statistical tests are confusing and present unnecessary information (values deviated from expectation; residuals; test statistics calculated from sample data during the hypothesis test...)

-I strongly suggest that the authors read and adapt the article to the STROBE-Vet guideline. This must be submitted as supplementary material.

https://strobevet-statement.org/

-The study area must be described in detail.

-The studied population should also be better described. What are its characteristics? What is the external validity of the results?

- The authors only mention that "targeted dog owners via websites, social media channels and newsletters directed at dog owners". Such a process should be better described. Based on what has been presented, the representativeness of the sample and the possibility of selection bias cannot be assessed.

6. PLOS authors have the option to publish the peer review history of their article (what does this mean?). If published, this will include your full peer review and any attached files.

Reviewer #1: No

Reviewer #2: No

---

## [Author Response · Author response to Decision Letter 0]

28 Mar 2020

We express our thanks for the constructive comments by the reviewers and yourself. In this letter we explain how we processed these comments and address the issues logically and transparently in the manuscript. We also indicate below where in the manuscript the key revisions can be found. 

Additional requirements: 

We have adjusted the manuscript according to the PLOS ONE style templates, including those for file naming. Also, we added S1 Appendix with the questionnaire items and provide the data as Supporting Information File - Data. 

Reviewer #1: 

1. Questionnaire and statistical approach

We now provide the questionnaire items in S1 Appendix. 

The reviewer's suggestion is to use more sophisticated statistics to explain our found variation in owner responses (i.e. by multivariate analysis). However, our approach was never to maximally explain the variation in response variates of interest by a preselected convenience set of explanatory factors. The study was not set-up for this, meaning the factors that we could test for associations would be arbitrary and not represent a logical set of candidate explanatory variables. Instead, we wanted to test a set of specific relationships and opted for a simple and straightforward statistical analysis by Chi squares, which suits the (limited) sample sizes. Despite our aim not being to test the effect of demographic variables, we still describe these variables to provide some insight on the participants to our convenience sample. We have added details on the collection of the demographic data in the Methods section (lines 136-139) and on the reason for the inclusion of thereof (lines 128-129).

The choice to collapse five-point Likert scales into binary categories was made for ‘satisfaction with dog ownership’ as answers here were skewed towards (very) satisfied, as made transparent in lines 144-146. Here we follow the approach that we used earlier (Van Herwijnen et al., 2018). The other measurement that we present as binary to our readers regards unwanted behaviour, which in our view is most optimally presented to them as ‘relevant or not’ to the factors we investigated (details are given in the Methods section lines 160-163). The reduction of scales may lead to the loss of some details but produces more reliable statistical outcomes, by avoiding low cell counts, and it fits with categorizations that would be meaningful to people.

The reviewer asked how the 491 entries were selected from the total study sample. These 491 entries are the male dogs in the combined dataset of male and female dogs, which was cleaned as described in lines 177-182. We excluded owners who left crucial questions unanswered, being the dog’s sex and reproductive status and (in the case of castrated dogs) whether or not correcting behaviour was a reason for castration. Furthermore, we excluded owners who indicated that they take care of the dog less than 50% of the time, as they may not have been the primary decision maker regarding the dog’s reproductive status. We also excluded owners whose dog was already castrated at acquisition, as these owners were likely uninvolved in the decision to castrate. Lastly, we excluded owners who had their dog castrated chemically, as we were unable to accurately assess the effect of the chemical castration. As stated above, this was made transparent in lines 177-182

2. Grammar

We adapted the grammar issues, such as regarding ‘assumingly’ for which we thank the reviewer for pointing these out. 

3. Specific knowledge gap - Introduction

We added lines 52-55 as to clarify the specific knowledge gap filled by this study. Also, we reworded the sentence on deciding to castrate. 

4. Questionnaire items and collection of demographic data

We added S1 Appendix, listing our questionnaire items and provided details on demographic data collection in the Methods section (lines 136-139).

5. Discussion

We acknowledge the effects of online self-selection and now provide more details on this in lines 449-464 and thank the reviewer for the interesting reference which we incorporated in our manuscript. 

Reviewer #2: 

1. Introduction

We thank the reviewer for pointing out these recent references and have added these to our manuscript. 

2. Sample size and power

We added details on our sample size and power calculation to the Methods section (lines 196-201). 

3. Questionnaire 

The questionnaire is now available as S1 Appendix and information on the development of the questionnaire can be found in lines 117-119 of the Methods section. 

4. The sample’s reflection of the dog (owning) population in The Netherlands

Here, the reviewer addresses a very interesting point. The meaningful comparison would be between our study population and the population of Dutch dog owners. Previously, our group tried to find information on the population of Dutch dog owners, e.g. by contacting Statistics Netherlands (CBS), but was not able to find useful facts. We mention the hiatus in the Methods section (lines 126-128). Furthermore, in the Discussion section we emphasize that our findings should not be extrapolated to the general population of dog owners without taking into account the limitations of our study (lines 465-466 and see also lines 449-451). 

5. Statistical analysis

We present the P-values in the text and the Tables for each studied comparison. With the Chi-square tests, we present standardized residuals to identify the cells with the largest contribution to the Chi-square test results. We mark residuals |>2| bold as this threshold is commonly accepted as a sufficiently large deviation between observed and expected values (Sharpe, 2015).

6. Results presentation

Indeed, Table 3 presents data on owners’ opinions regarding castration affecting male dog behaviour at a population level. Readers will see that distributions for these opinions differ between owners of castrated and intact dogs. The differences in opinions regarding the population level or the own dog are discussed in the Discussion section (lines 432-442). We have specified more clearly that Table 3 regards opinions at a population level, not the reported effect for the own dog. We have adapted all Tables, so they now present proportions as percentages.

7. Causal inference

We have made more salient in the Discussion section that our study was not set up for making casual inferences (lines 449-451). Our aim was to provide insight into owner received professional advice, opinions on castrating male dogs and behavioural reasons for this decision. We feel these findings indicate the need for prospective study set ups that can provide causal evidence as the reviewer justly points out and we have stressed this need in our manuscript (lines 469-470; 478-480). 

8 & 9. Table 4 and owner perception

We have reworded the text hoping to provide more clarity while still reporting our results accurately. Also, we agree with the reviewer that our data may reflect owner perception and consequentially this is one of the main points of the Discussion section (see for example the text starting at line 414). 

Reviewer #3: 

1. Introduction and discussion

We have shortened and simplified the Introduction and Discussion. 

2. Tables and results of statistical tests

We present the P-values in the text and Tables for each comparison. With the Chi-square tests, we present standardized residuals to identify the cells with the largest contribution to the Chi-square test results. We mark residuals |>2| bold as this threshold is commonly accepted as a sufficiently large deviation between observed and expected values (Sharpe, 2015). We have adapted all Tables, so they now present proportions as percentages.

3. STROBE-Vet guidelines

We thank the reviewer for pointing us to the STROBE-Vet guidelines and have applied them where this was possible, as our study regards questionnaire-based research and was not an observational study. Also, we improved recognition of key elements, such as through using words as ‘limitations’ (line 30, 449 and 466) to indicate these sections. Changes made include the direct mentioning of study type in the abstract (line 15) and the mentioning of study limitations (Abstract lines 30-31 and Discussion lines 449-451). We also added a sentence on the lack of conflicts of interests (line 491).

4. Study area

The study area has been detailed further (lines 46-55). 

5. Study population

The characteristics of the study population can be found in lines 223-237 and the Discussion provides information on representativeness and external validity of the results (lines 449-451; 465-466) and on selection effects in lines 451-464. In the Methods, we have also provided more details on how the dog owners were targeted (lines 124-126). 

With these adaptations made to our manuscript, we expect to have addressed all points raised by the reviewers.

---

## [Decision Letter · Decision Letter 1]

12 May 2020

PONE-D-20-00127R1

Self-reports of Dutch dog owners on received professional advice, their opinions on castration and behavioural reasons for castrating male dogs

PLOS ONE

Dear Dr Pascalle Elise Maria Roulaux

Thank you for submitting your manuscript to PLOS ONE. After careful consideration, we feel that it has merit but does not fully meet PLOS ONE’s publication criteria as it currently stands. Therefore, we invite you to submit a revised version of the manuscript that addresses the points raised during the review process.

Many thanks for resubmitting your manuscript to PLOS One

It was reviewed by the same reviewers as previously, and they have suggested further changes to the manuscript prior to acceptance

If you could write a detailed response to reviewers, that will expedite review when resubmitted.

I wish you the best of luck with your changes

Hope you are keeping safe and well in this difficult time

Thanks

Simon

We would appreciate receiving your revised manuscript by Jun 26 2020 11:59PM. To enhance the reproducibility of your results, we recommend that if applicable you deposit your laboratory protocols in protocols.io, where a protocol can be assigned its own identifier (DOI) such that it can be cited independently in the future. For instructions see: http://journals.plos.org/plosone/s/submission-guidelines#loc-laboratory-protocols

We look forward to receiving your revised manuscript.

Kind regards,

Simon Russell Clegg, PhD

Academic Editor

PLOS ONE

Reviewers' comments:

Reviewer's Responses to Questions

**Comments to the Author**

1. If the authors have adequately addressed your comments raised in a previous round of review and you feel that this manuscript is now acceptable for publication, you may indicate that here to bypass the “Comments to the Author” section, enter your conflict of interest statement in the “Confidential to Editor” section, and submit your "Accept" recommendation.

Reviewer #2: (No Response)

Reviewer #3: All comments have been addressed

2. Is the manuscript technically sound, and do the data support the conclusions?

Reviewer #2: Partly

Reviewer #3: (No Response)

3. Has the statistical analysis been performed appropriately and rigorously? 

Reviewer #2: No

Reviewer #3: (No Response)

4. Have the authors made all data underlying the findings in their manuscript fully available?

Reviewer #2: Yes

Reviewer #3: (No Response)

5. Is the manuscript presented in an intelligible fashion and written in standard English?

Reviewer #2: Yes

Reviewer #3: (No Response)

6. Review Comments to the Author

Reviewer #2: The authors made an immense effort to modify the manuscript to take care of my comments. Although I still cannot entirely agree with some statements and conclusions, I understand that we got into the difficult line between reviewer's necessity and authors' latitude to decide the way they want to show their results to the community. Therefore, I will only maintain two points that I still think are necessary for eventual publication in PLOS One.

1) Regarding sample size, I do not think the text included is neither sufficient nor correct. Authors should be able to calculate the power of the study for detecting, for instance, a difference in a certain proportion, say 10%. Please provide this using standard formula, not references for average differences.

2) Regarding the presentation of results, there was no improvement. In this type of study, in which you have two groups (castrated vs. intact) and proportions as responses, you need a better approach to present results.

As I already mentioned, there is no reason to present deviations from expected values; we just need p-values for each comparison. Table 1 could be seen as three 2-by-2 tables comparing advised or not by a certain professional and being castrated or not. For instance, we can see that 81% of owners of castrated dogs received advice from veterinarians against only 63% of owners of intact dogs, is this difference statistically significant? The same question applies to therapists and trainers. Three comparisons, three p-values, that's all we need. There is no need for presenting deviations, the value of the chi-square or the df. The same applies to all tables.

Reviewer #3: (No Response)

7. PLOS authors have the option to publish the peer review history of their article (what does this mean?). If published, this will include your full peer review and any attached files.

Reviewer #2: No

Reviewer #3: No

---

## [Author Response · Author response to Decision Letter 1]

27 May 2020

Responses to the editor and reviewer are added as separate cover letters.

---

## [Editor Report · Decision Letter 2]

5 Jun 2020

Self-reports of Dutch dog owners on received professional advice, their opinions on castration and behavioural reasons for castrating male dogs

PONE-D-20-00127R2

Dear Dr Pascalle Elise Maria Roulaux

We’re pleased to inform you that your manuscript has been judged scientifically suitable for publication and will be formally accepted for publication once it meets all outstanding technical requirements.

Kind regards,

Simon Clegg, PhD

Academic Editor

PLOS ONE

Additional Editor Comments (optional):

Many thanks for resubmitting your manuscript to PLOS One

As you have addressed all the reviewers comments, and two of the reviewers are in agreement to accept, I have recommended the manuscript for publication

You should hear from the Editorial Office soon

It was a pleasure working with you, and I wish you the best of luck for your future research

Hope you are keeping safe and well in these difficult times

Thanks

Simon

---

## [Editor Report · Acceptance letter]

11 Jun 2020

PONE-D-20-00127R2 

Self-reports of Dutch dog owners on received professional advice, their opinions on castration and behavioural reasons for castrating male dogs 

Dear Dr. Roulaux:

I'm pleased to inform you that your manuscript has been deemed suitable for publication in PLOS ONE. Congratulations! Your manuscript is now with our production department. 

Kind regards, 

on behalf of

Dr. Simon Clegg 

Academic Editor

PLOS ONE